# PR-619, a General Inhibitor of Deubiquitylating Enzymes, Diminishes Cisplatin Resistance in Urothelial Carcinoma Cells through the Suppression of c-Myc: An In Vitro and In Vivo Study

**DOI:** 10.3390/ijms222111706

**Published:** 2021-10-28

**Authors:** Fu-Shun Hsu, Wei-Chou Lin, Kuan-Lin Kuo, Yen-Ling Chiu, Chen-Hsun Hsu, Shih-Ming Liao, Jun-Ren Dong, Shing-Hwa Liu, Shih-Chen Chang, Shao-Ping Yang, Yueh-Tang Chen, Ruei-Je Chang, Kuo-How Huang

**Affiliations:** 1Graduate Institute of Clinical Medicine, National Taiwan University College of Medicine, Taipei 100, Taiwan; fs_hsu@outlook.com (F.-S.H.); yenling.chiu@gmail.com (Y.-L.C.); 2Department of Urology, YangMing Branch of Taipei City Hospital, Taipei 111, Taiwan; 3Department of Exercise and Health Sciences, University of Taipei, Taipei 111, Taiwan; 4Department of Food and Beverage Management, Vanung University, Taoyuan 320, Taiwan; 5Department of Urology, National Taiwan University Hospital, National Taiwan University College of Medicine, Taipei 100, Taiwan; antibody0123@gmail.com (K.-L.K.); chanhsun.hsu@gmail.com (C.-H.H.); sanguine444@gmail.com (S.-M.L.); s8907001166@gmail.com (J.-R.D.); emilyang719@gmail.com (S.-P.Y.); b08401132@ntu.edu.tw (Y.-T.C.); fenoxycarb@gmail.com (R.-J.C.); 6Department of Pathology, National Taiwan University Hospital, National Taiwan University College of Medicine, Taipei 100, Taiwan; weichou8@ms52.hinet.net; 7Graduate Institute of Toxicology, National Taiwan University College of Medicine, Taipei 100, Taiwan; shinghwaliu@ntu.edu.tw; 8Department of Medical Research, Far Eastern Memorial Hospital, New Taipei City 220, Taiwan; 9Graduate Institute of Medicine and Graduate Program in Biomedical Informatics, Yuan Ze University, Taoyuan 320, Taiwan; 10Graduate Institute of Immunology, National Taiwan University College of Medicine, Taipei 100, Taiwan; b101103043@tmu.edu.tw

**Keywords:** deubiquitylating enzymes, urothelial carcinoma, chemoresistance

## Abstract

Cisplatin-based chemotherapy is the standard treatment for bladder urothelial carcinoma (UC). Most patients experience chemoresistance, the primary cause of treatment failure, which leads to disease relapse. The underlying mechanism of chemoresistance involves reduced apoptosis. In this study, we investigated the antitumor effect of the deubiquitylating enzyme inhibitor PR-619 in cisplatin-resistant bladder UC. Deubiquitinase (ubiquitin-specific protease 14 (USP14) and USP21) immunohistochemical staining demonstrated that deubiquitination is related to chemoresistance in patients with metastatic UC and may be a target for overcoming chemoresistance. Cytotoxicity and apoptosis were assessed using fluorescence-activated flow cytometry and a 3-(4,5-dimethylthiazol-2-yl)-2,5-diphenyl tetrazolium assay, and PR-619 was found to enhance the cytotoxic and apoptotic effects of cisplatin in cisplatin-resistant T24/R cells. Mitigated cisplatin chemoresistance was associated with the concurrent suppression of c-Myc expression in T24/R cells. Moreover, the expression of c-Myc was upregulated in human bladder UC specimens from patients with chemoresistance. Experiments in a xenograft nude mouse model confirmed that PR-619 enhanced the antitumor effects of cisplatin. These results are promising for the development of therapeutic strategies to prevent UC chemoresistance through the combined use of chemotherapeutic agents/deubiquitination inhibitors (PR-619) by targeting the c-Myc pathway.

## 1. Introduction

Bladder urothelial carcinoma (UC) has been estimated to be the sixth most common cancer in the United States, accounting for more than 90% of bladder cancers, with an estimated 74,000 new cases in 2015 [1,2]. Almost 50% of high-grade and muscle-invasive bladder UC cases progress to the metastatic phase despite radical cystectomy. While the primary therapy for metastatic bladder UC is cisplatin-based chemotherapy, the prognosis for these patients is rather poor [3,4,5]. Despite a positive initial response to chemotherapy, most patients experience relapse and eventually die as a result of chemoresistance. Therefore, it is imperative to develop strategies to circumvent chemoresistance to improve the outcomes of metastatic bladder cancer [6]. For decades, researchers have attempted to understand the molecular mechanism underlying chemoresistance and develop strategies to overcome it in patients with various cancer types. Multiple pathways and genetic and epigenetic factors contribute to resistance to chemotherapy [6]. Almost all conventional chemotherapeutic drugs in clinical use induce apoptosis of cancer cells. Reduced apoptosis in response to chemotherapeutic drugs plays a crucial role in chemoresistance and is associated with the altered expression patterns of antiapoptotic and proapoptotic proteins in chemo-resistant cells [7]. Furthermore, the downregulation of the expression of proapoptotic proteins or upregulation of antiapoptotic molecules such as c-Myc are major events contributing to chemoresistance [8].

DNA-damaging agents are widely used in chemotherapy. Cisplatin (cis-diamminedichloroplatinum II), a platinum-based chemotherapeutic drug, is the most commonly used agent that crosslinks with the purine bases of DNA, leading to the formation of DNA adducts and the consequent inhibition of DNA replication and transcription [9]. Cancer cells respond to DNA damage through the activation of a network of damage response pathways that regulate cell cycle retardation, proliferation, DNA repair, and death [9]. The suppression of apoptosis following cisplatin-induced cytotoxic insult may be a major factor contributing to chemotherapy resistance [10,11,12,13]. Among the numerous factors associated with chemoresistance, reduced apoptosis of tumor cells is the most critical. Dysregulation of c-Myc gene expression induces apoptosis in various cells. The rate of cell apoptosis and its sensitivity to inducing factors depend on the concentration of the cellular Myc protein [14]. During apoptosis, the expression of c-Myc is highly upregulated. If antisense oligonucleotides are used to block c-Myc gene expression, apoptosis is severely disturbed. Once c-Myc is obstructed, the cell initiates an apoptotic program that leads to the formation of tumors [8,15].

The post-translational attachment of ubiquitin is a key determinant of protein fate. The ubiquitination process requires three enzymes, namely, ubiquitin-activating, ubiquitin-conjugating, and ubiquitin ligase enzymes. Ubiquitin can be conjugated to substrates involved in various cellular processes to target these proteins for proteasomal degradation [16]. Deubiquitinating enzymes (DUBs) remove covalently attached ubiquitin from proteins, thereby controlling substrate activity and/or abundance; therefore, the expression of these enzymes can be efficiently modulated to maintain cellular protein homeostasis [17]. Approximately 100 DUBs encoded in the human genome [17] have been identified as promising targets for cancer therapy [18,19]. 

To date, pharmacological studies have identified several novel DUB inhibitors or antagonists for potential clinical use [20]. The pan-DUB inhibitor PR-619 has a broad specificity and is known to inhibit multiple DUBs [21]. PR-619 has been reported as an effective treatment for some cancers [22,23] and appears to be a promising agent when used in combination with current chemotherapeutic drugs. However, the potential anti-chemoresistance effects of PR-619 on UC cells remain unclear. Therefore, we conducted in vitro and in vivo experiments to investigate the efficacy of PR-619 in the inhibition of human cisplatin-resistant UC cells. Moreover, we explored the cytotoxic effects of cisplatin plus PR-619 on cisplatin-resistant UC cells and investigated the underlying mechanisms.

## 2. Results

### 2.1. The Cisplatin-Resistant Human UC Cell Line (T24/R) Was Resistant to Treatment with Cisplatin In Vitro and In Vivo

The cisplatin-resistant UC cell line T24/R was established as described in our previous study [24]. We assessed the apoptotic effect of cisplatin on T24 and T24/R cells using Annexin V and propidium iodide (PI) staining with flow cytometry. As shown in Figure 1A, the T24/R cell line was more resistant to cisplatin than the T24 cell line in vitro. We then assessed the inhibitory effect of cisplatin on T24 or T24/R xenograft nude mice. Consistent with the in vitro results, the growth of the T24-xenografted tumors was significantly inhibited by cisplatin compared to that of the tumors from the control group (saline) at 28 days. However, T24/R xenografts were resistant to cisplatin treatment (Figure 1B–D). 

### 2.2. USP21 Overexpression Is Associated with Chemoresistance in Patients with Chemoresistant UC

To clarify whether deubiquitination can be a target for the treatment of chemoresistant urothelial cancer, we analyzed the expression of USP14 and USP21 in samples from patients with bladder UC with different progressions (five chemosensitive and five chemoresistant patients who had received systemic chemotherapy with gemcitabine/cisplatin) using IHC staining. USP14 and USP21 are important members of the DUB group and are potential contributors to cell proliferation, metastasis, and chemoresistance in various malignancies [25,26,27]. Some studies have also found that USP14 and USP21 could function as oncogenes in bladder cancer [28,29,30,31]. As shown in Figure 2A,B, the immunoreactivity of these DUBs was stronger in chemoresistant tumors (bottom) than in chemosensitive UC tumors (top). The high expression of DUBs (USP14 and USP21) was associated with resistance to chemotherapy in patients with UC. These results show that deubiquitination could be a potential target for the treatment of chemoresistant UC. The use of PR-619 as a pan-DUB inhibitor can potentially diminish chemoresistance in UC by inhibiting deubiquitination.

### 2.3. PR-619 Effectively Induced Viability Inhibition, Apoptosis, and Endoplasmic Reticulum Stress-Related Apoptosis in T24/R Cells

Next, we investigated the antitumor effects of PR-619 on cisplatin-resistant UC cells (T24/R). As shown in Figure 3A, PR-619 treatment significantly inhibited cell viability at 24 and 48 h in a dose-dependent manner. Furthermore, PR-619 (20 μM) treatment for 48 h significantly induced apoptosis in cisplatin-resistant T24/R cells (Figure 3B). We observed an increase in the protein levels of ER stress-related apoptosis markers, caspase-4 and phospho-JNK, after PR-619 treatment (Figure 3C). We also examined the effect of PR-619 on T24/R cell cycle progression by using flow cytometry. The results showed that 20 μM PR-619-treated T24/R cells showed G2/M phase arrest after 48 h (Figure 3D). In addition, the expression of phospho-p53, p21, and p27 (cyclin-dependent kinase inhibitors) increased 48 h after PR-619 treatment (Figure 3E).

### 2.4. PR-619 Enhanced the Cytotoxic Effects of Cisplatin on T24/R Cells, and the Attenuation of Cisplatin Resistance Was Related to the Concurrent Suppression of c-Myc

We evaluated the cytotoxic effects of PR-619 in combination with cisplatin on T24/R cells. The combined effects of PR-619 and cisplatin on T24/R cells following treatment for 48 h were analyzed using the MTT assay. PR-619 significantly enhanced cisplatin-induced cytotoxicity (Figure 4A). Western blotting analysis for cleaved caspase-3 and c-Myc expression demonstrated the PR-619-mediated suppression of cisplatin-induced c-Myc, Bcl2, and phospho-Bcl2 expression upregulation, which potentiated cisplatin-induced apoptosis (Figure 4B). We hypothesized that upregulated c-Myc and Bcl2 levels were associated with cisplatin resistance in T24/R cells. PR-619 diminished cisplatin resistance in T24/R cells and re-sensitized them to cisplatin, consistent with the downregulation of c-Myc and Bcl2 expression. 

### 2.5. Upregulation of c-Myc Is Associated with Chemoresistance in Patients with Chemoresistant UC

We compared the expression of USP21 in chemosensitive and chemoresistant UC tumors (five chemosensitive and four chemoresistant patients) using IHC staining, as shown in Figure 5. In our previous study, we had demonstrated higher expression levels of c-Myc in chemoresistant UC cells than in chemosensitive UC cells [32].

### 2.6. PR-619 Enhanced the Antitumor Effect of Cisplatin in a Xenograft Mouse Model of T24/R 

We evaluated the antitumor effects of PR-619 alone and in combination with cisplatin in a xenograft mouse model. A T24/R cells/Matrigel mixture was subcutaneously injected into homozygous nude mice. The mice were divided into four groups as follows: saline (untreated control, *n* = 5), cisplatin (*n* = 6), PR-619 (*n* = 7), and cisplatin/PR-619 combination (*n* = 8). Mice were intraperitoneally administered different regimens for 36 days. The combination of PR-619 and cisplatin showed the highest antitumor effect as compared with cisplatin or PR-619 treatment alone on T24/R xenografts (Figure 6A,B). 

## 3. Discussion

Cisplatin has been used to treat bladder cancer since 1978 [33] and remains the primary constituent of standard chemotherapeutic regimens. However, most patients with metastatic UC rapidly become unresponsive to chemotherapy, which results in treatment failure. The emerging chemoresistance leads to ominous prognoses. The precise molecular mechanisms underlying cisplatin resistance remain unclear, and novel antitumor agents or new drug combinations are urgently required [13,34]. 

The c-Myc gene is one of the principal members of the Myc gene family. c-Myc functions as a transcription factor that activates expression of several other genes required for cell cycle progression, proliferation, and apoptosis. The knockdown of c-Myc expression suppresses the self-renewal, tumorigenicity, invasive ability, and drug resistance of cancer stem cells in numerous malignancies [14,35,36]. Our study revealed that PR-619 enhanced cisplatin-induced cytotoxicity and alleviated cisplatin resistance in cisplatin-resistant T24/R cells and concurrently suppressed c-Myc expression. Treatment with PR-619 alone or in combination with cisplatin enhanced the activation of caspases and downregulated c-Myc expression. The IHC staining of clinical specimens from patients with chemoresistant metastatic UCs also revealed the overexpression of c-Myc. These findings highlight the potential clinical benefit of targeting c-Myc to circumvent chemotherapy resistance.

Chemoresistance may involve a variety of internal cellular processes, including drug efflux, drug inactivation, drug target changes, epithelial–mesenchymal transition, inherent cell heterogeneity, epigenetic effects, or their combinations. DUBs are promising targets for cancer treatment. USP14 and USP21 can inhibit proteasomes in vitro and protein turnover in cells. Previous studies have demonstrated that USP21 and USP14 were upregulated in bladder carcinoma and played an important role in tumor cell proliferation and metastasis [28,35]. 

In this study, we demonstrated the overexpression of USP14 and USP21 in chemoresistant UCs. These were the preliminary efforts to understand the functional relevance of USP14 and USP21 in chemoresistance and suggest that USP14 and USP21 might not only be involved in UC tumorigenesis and metastasis, but also in the chemoresistance of UCs. We found that c-Myc protein levels significantly increased along with USP14 and USP21. These observations support the hypothesis that low expression of USP14 and USP21 in UCs may be correlated with chemotherapy responses, and that USP14 and USP21 may potentially serve as biomarkers to predict chemotherapy responses. 

PR-619 as a pan-DUB inhibitor effectively induced cytotoxicity and apoptosis in chemoresistant UC cells in vitro and inhibited the growth of chemoresistant UC tumors in vivo. PR-619 circumvented cisplatin resistance in UCs and concurrently suppressed c-Myc expression. The exact mechanism by which USP14 and USP21 influence tumorigenesis and drug resistance remains unknown. Nevertheless, pharmacological studies of DUBs have identified specific DUB inhibitors for potential clinical use. 

In summary, PR-619 enhanced the antitumor effect of cisplatin and conquered drug resistance in cisplatin-resistant UC cells, which could be regulated by the downregulation of c-Myc. These results are crucial for the clinical application of PR-619 and the identification of chemosensitizers to augment and improve the therapeutic efficacy of treatments for cisplatin-resistant UC cells.

## 4. Materials and Methods

### 4.1. Cell Culture

The T24 cell line was purchased from the Bioresource Collection and Research Center (Hsinchu, Taiwan). The cisplatin-resistant UC cells (T24/R) were derived from the original parental T24 cells as described in our previous studies [24]. The two cell lines were cultured in RPMI-1640 containing 10% fetal bovine serum at 37 °C with 5% CO_2_. All the materials for cell culture were obtained from Invitrogen (Carlsbad, CA, USA). 

### 4.2. Chemicals and Antibodies

PR-619 was obtained from MedChemExpress (Junction, NJ, USA) and cisplatin from Merck Millipore (Billerica, MA, USA). All other chemicals were purchased from Sigma-Aldrich (St. Louis, MO, USA) or Merck Millipore. For Western blot analysis, antibodies against cleaved caspase-3 (#9661), cleaved caspase-8 (#9496), B cell lymphoma (Bcl)-2 (#15071), phospho-Bcl-2 (#2824), phospho-p53 (#9284), caspase-4 (#4450), phospho-Janus kinase (JNK) (#9255), and c-Myc (#18683) were procured from Cell Signaling Technology (Danvers, MA, USA). The antibodies against β-actin (#109) and glyceraldehyde 3-phosphate dehydrogenase (GAPDH, #100118) were supplied by GeneTex (Irvine, CA, USA); those against α-tubulin (sc-5286), P21 (sc-6264), P27 (sc-1641), and JNK (sc-571) were obtained from Santa Cruz Biotechnology (Santa Cruz, CA, USA). For IHC, the USP14 antibody (#MA5-32821) was purchased from Invitrogen, the USP21 antibody (#17856-1-AP) was purchased from Proteintech (Chicago, IL, USA), and the c-Myc (#ab32072) antibody was purchased from Abcam (Cambridge, UK). 

### 4.3. Measurement of Cell Viability

Cell viability was analyzed using the 3-(4,5-dimethylthiazol-2-yl)-2,5-diphenyl tetrazolium bromide (MTT) assay (Sigma-Aldrich). In brief, cells were seeded into wells of a 96-well plate (5000 cells/well) and incubated at 37 °C for 24 h. Then, the cells were subjected to various treatments for indicated time periods and incubated in a medium containing MTT (0.5 mg/mL) at 37 °C for 4 h. The reduced crystals were dissolved in dimethyl sulfoxide (DMSO), and the absorbance was measured at a 570 nm wavelength using the Multiskan GO Microplate Spectrophotometer (Thermo Fisher Scientific, Rockford, IL, USA) [36].

### 4.4. Western Blot Analysis

Western blot analysis was performed according to the methods described in previous studies [36,37,38]. The final antibody-labeled membranes were visualized using the ImageQuant LAS 4000 system (GE Healthcare, Uppsala, Sweden). 

### 4.5. Apoptosis Assay

Cellular apoptosis assay was performed with the Muse Annexin V and Dead Cell Kit (Merck Millipore) according to the manufacturer’s protocol as previously described [24,38]. Apoptotic cells were detected and quantified using Muse Cell Analyzer flow cytometry (Merck Millipore).

### 4.6. Cell Cycle Analysis 

Cells were seeded, cultured to 40% confluence, and then treated with DMSO (control) or PR-169 for 48 h. The cell cycle population was analyzed, as previously described, using the Muse Cell Cycle Assay kit with a Muse Cell Analyzer flow cytometer (Merck Millipore) [24].

### 4.7. Immunohistochemical (IHC) Staining in Human UC Specimens

Formalin-fixed and paraffin-embedded tissue blocks were collected from patients with chemosensitive bladder UC or metastatic bladder UC who had received systemic chemotherapy with gemcitabine and cisplatin. Based on disease progression or responsiveness during chemotherapy, three patients were classified as chemoresistant and three as chemosensitive. IHC staining with USP14, USP21, and c-Myc antibodies was performed on 5 μm sections of formalin-fixed paraffin-embedded specimens, as previously described [39]. The concentrations of antibodies we used for IHC staining were USP14 antibody: 3 μg/mL, USP21 antibody: 5 μg/mL, and c-Myc antibody: 5 μg/mL. A board-certified pathologist (W.-C.L.) who was blinded to the clinical data assessed the c-Myc immunoreactivity. The staining intensity was categorized as 0 (negative), 1 (weakly positive), 2 (moderately positive), and 3 (strongly positive). The mean percentage of positively stained tumor cells was determined by counting at least 10 random fields in each section. IHC scores were calculated as the four-tiered intensity (0, 1+, 2+, 3+) of tumor cells multiplied by the mean percentage of the positive staining of tumor cells. The Research Ethics Committee B of the National Taiwan University Hospital approved the study protocol and waived the need for informed consent for the use of existing biosamples (201901032RINB). The committee operates in accordance with the Good Clinical Practice guidelines and governmental laws and regulations.

### 4.8. In Vivo Xenograft Experiments of PR-619 and Cisplatin Combination

T24/R cells (6 × 10^5^) were suspended in saline and mixed with Matrigel at an equal volume (BD, Franklin Lakes, NJ, USA). The mixture was subcutaneously injected into 6 to 8 week-old nude mice (BALB/cAnN.Cg-Foxnlnu/CrlNarl) obtained from the Taiwan National Laboratory Animal Center. The mice were divided into four groups as follows: saline (untreated control, *n* = 5), cisplatin (*n* = 6), PR-619 (*n* = 7), and cisplatin/PR-619 combination (*n* = 8). Once the tumor volumes reached 150–200 mm^3^, mice were intraperitoneally treated with PR-619 (10 mg/kg each day), cisplatin (5 mg/kg, twice/week), or cisplatin in combination with PR-619 for 36 days. Tumor dimensions were measured every 4 days with calipers, and tumor volume was calculated as follows: longest tumor diameter × (shortest tumor diameter)^2^/2. On the final day, the tumors were excised from the mice and photographed. The animal experiments complied with the Animal Research: Reporting of In Vivo Experiments guidelines and were approved by the National Taiwan University College of Medicine and the College of Public Health’s Institutional Animal Care and Use Committee (No. 20190194) [36]. 

### 4.9. Statistical Analysis

GraphPad Prism 6 software (GraphPad Software, San Diego, CA, USA) was used for statistical analyses. The results are expressed as the mean ± standard deviation or standard error of the mean. Data from two groups were analyzed using the two-tailed Student’s *t*-test. Data from multiple groups were analyzed using one-way analysis of variance followed by the Bonferroni post-hoc test. Statistical significance was set at *p* < 0.05.

## 5. Conclusions

PR-619 enhanced the antitumor effect of cisplatin, alleviated drug resistance in cisplatin-resistant UC cells, and suppressed c-Myc expression. The overexpression of USP14 and USP21 in chemoresistant UCs correlated with drug resistance. Our results demonstrate a potential therapeutic strategy to conquer chemoresistance in UC. Future studies should focus on finding novel small-molecule inhibitors of USP14 and USP21 and investigating their therapeutic effects.

## Figures and Tables

**Figure 1 ijms-22-11706-f001:**
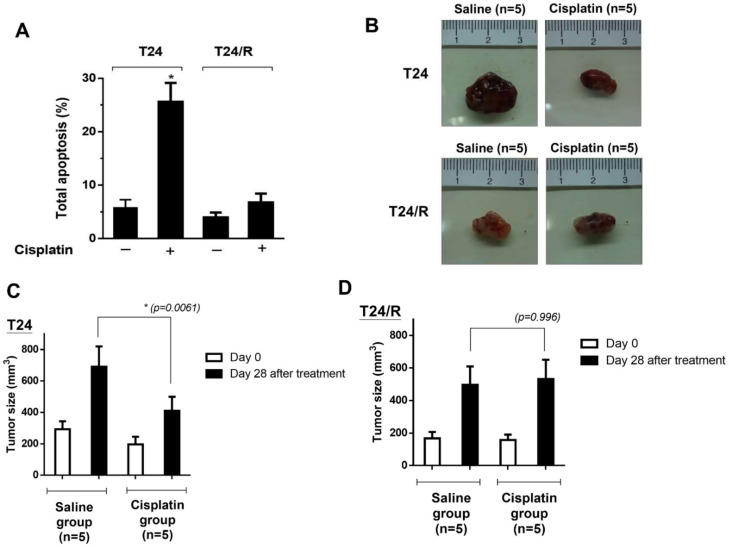
Cisplatin-induced apoptosis was mitigated in cisplatin-resistant urothelial carcinoma cell lines. (**A**) Cells were exposed to cisplatin (15 μM) and dimethyl sulfoxide for 48 h. Apoptotic cells were analyzed using fluorescence-activated cell sorting flow cytometry with propidium iodide and annexin V-FITC staining. Data are presented as means ± standard deviation (SD); * *p* < 0.05 represents a significant difference between the indicated groups. (**B**–**D**) The in vivo xenograft model demonstrated that T24/R cells were resistant to cisplatin. The T24 and T24/R tumor-bearing mice were divided into: saline as untreated control, (*n* = 5), and cisplatin-treated (5 mg/kg, *n* = 5) twice per week for 28 days, respectively. Tumor volumes were measured before treatment and 28 days after treatment. The data are presented as means ± SD; * *p* < 0.05 represents a significant difference between the indicated groups.

**Figure 2 ijms-22-11706-f002:**
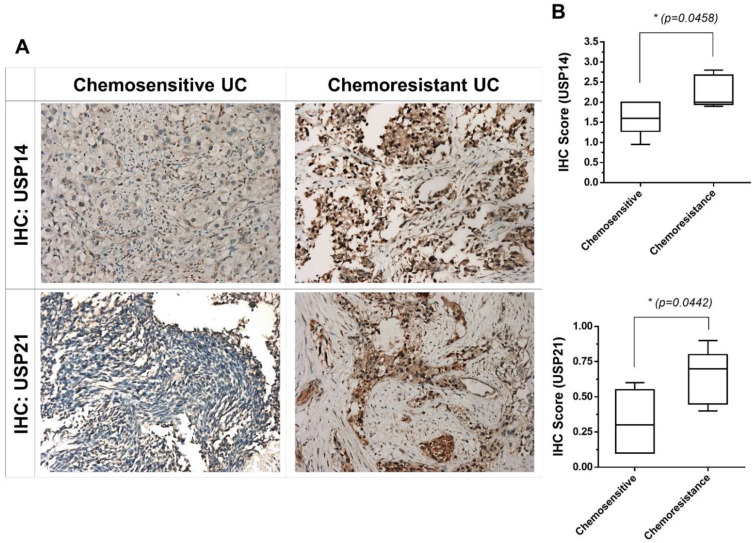
Immunohistochemical staining of USP14 and USP21 in tumor tissues obtained from patients with chemosensitive and chemoresistant metastatic bladder urothelial carcinoma. (**A**) Five chemoresistant and five chemosensitive clinical tumor samples were IHC-stained with USP14 and USP21 antibodies. The tissue sections were photographed at 200× magnification. (**B**) The IHC scores of USP14 and USP21 expression from chemoresistant and chemosensitive tumors. Data were analyzed using the unpaired two-tailed Student’s *t*-test and are presented as means ± standard deviation; * *p* < 0.05 represents a significant difference between the indicated groups.

**Figure 3 ijms-22-11706-f003:**
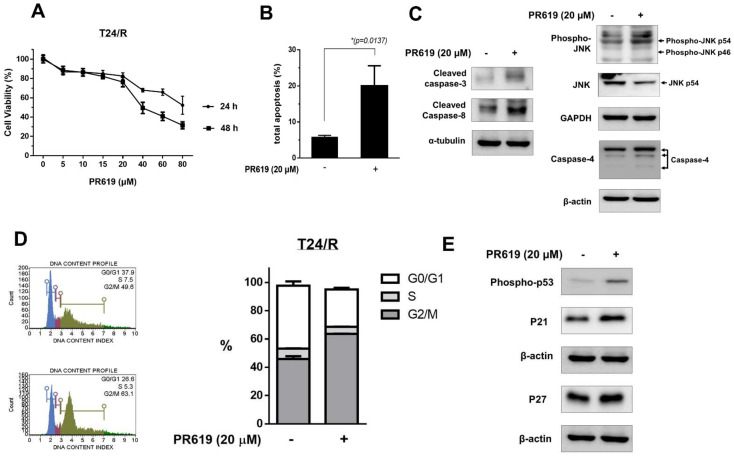
PR-619 effectively induces cytotoxicity, apoptosis, endoplasmic reticulum stress-related apoptosis, and cell cycle arrest in cisplatin-resistant human urothelial carcinoma (UC) cells (T24/R). (**A**) Cisplatin-resistant UC cell lines (T24/R) received mock treatment (dimethyl sulfoxide; DMSO) or various concentrations of PR-619 (10–45 μM) for 48 h. Cell viability was assessed using the MTT assay. (**B**) T24/R cells were treated with PR-619 (20 μM) or DMSO for 48 h. Apoptotic cells were analyzed using fluorescence-activated cell sorting with propidium iodide and Annexin V-FITC staining. Data are presented as means ± standard deviation; * *p* < 0.05 represents a significant difference between the indicated groups. (**C**) Cell lysates were harvested and Western blotting was performed with specific antibodies for the stress-related molecules, phospho stress-activated protein kinase/c-Jun N-terminal kinase (Thr183/Tyr185), and the ER stress-related apoptosis molecule caspase-4. The results are representative of at least three independent experiments. (**D**) PR-619 induces G2/M arrest in cisplatin-resistant human UC cells (T24/R). T24/R cells were treated with PR-619 (20 μM) or DMSO for 48 h. The cell cycle progression was analyzed using flow cytometry with propidium iodide staining. Quantitative data are presented as means ± SD of independent experiments. * *p* < 0.05 was considered significant as compared with the control. (**E**) T24/R cells were treated with PR-619 (20 μM) or DMSO for 48 h. The expression of cyclin-dependent kinase inhibitors p21 and p27 in total cell lysates was analyzed using Western blotting. The results are representative of at least three independent experiments.

**Figure 4 ijms-22-11706-f004:**
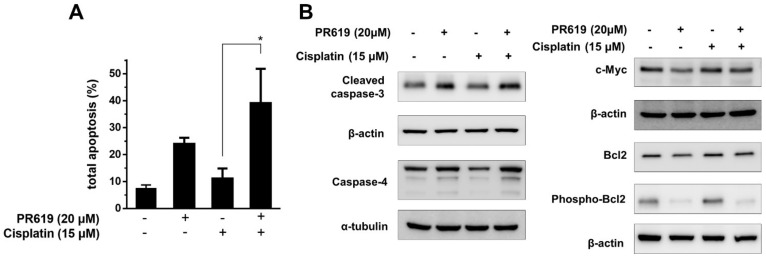
The antitumor effects of cisplatin in T24/R cells are enhanced by PR-619. The alleviation of cisplatin resistance was associated with the concurrent suppression of c-Myc expression. (**A**) Cells were exposed to cisplatin (15 μM) or PR-619 (20 μM) alone or in combination for 48 h. Apoptotic cells were analyzed using FACS with propidium iodide and Annexin V-FITC staining. (**B**) T24/R cells were treated with cisplatin (15 μM) or PR-619 (20 μM) alone or in combination for 48 h. Cell lysates were subjected to Western blotting analysis to detect cleaved caspase-3, c-Myc, Bcl2, and phospho-Bcl2. Values from the quantitative analyses of apoptosis are presented as the means ± standard deviation; * *p* < 0.05 represents a significant difference between the indicated groups.

**Figure 5 ijms-22-11706-f005:**
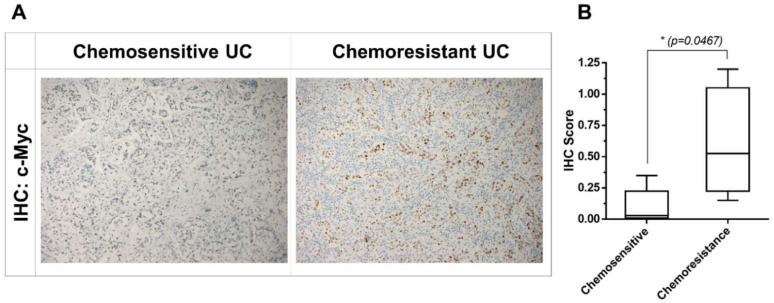
Immunohistochemical staining of c-Myc in tumor tissues obtained from patients with chemosensitive and chemoresistant metastatic bladder urothelial carcinoma. (**A**) Five chemosensitive and four chemoresistant clinical tumor samples were IHC-stained with c-Myc antibodies. The tissue sections were photographed at 100× magnification. (**B**) IHC scores of c-Myc expression from chemoresistant and chemosensitive tumors. Data were analyzed using unpaired two-tailed Student’s *t*-test, and values are presented as means ± standard deviation; * *p* < 0.05 represents a significant difference between the indicated groups.

**Figure 6 ijms-22-11706-f006:**
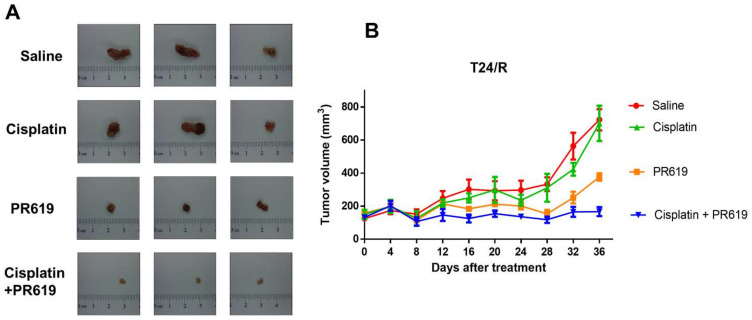
PR-619 enhances the antitumor effect of cisplatin in mice with xenografts of cisplatin-resistant urothelial carcinoma (UC) cells (T24/R). Nude mice with cisplatin-resistant T24/R UC xenograft tumors were treated with saline (untreated control, *n* = 5), cisplatin (*n* = 6), PR-619 (*n* = 7), or cisplatin/PR-619 combination (*n* = 8) for 36 days. (**A**) The tumor images represent the excised tumors from each group. (**B**) Tumor volume of each group during the 36-day treatment period. The data are presented as means ± standard error of the mean.

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
