# Peer review of "PR-619, a General Inhibitor of Deubiquitylating Enzymes, Diminishes Cisplatin Resistance in Urothelial Carcinoma Cells through the Suppression of c-Myc: An In Vitro and In Vivo Study"

_ijms, 2021, doi:10.3390/ijms222111706_

Round 1
Reviewer 1 Report
The authors investigated the antitumor effect of the deubiquitylating enzyme in-30 inhibitor PR-619 in cisplatin-resistant bladder UC. The study can be improved by correcting figure legends and providing details in materials and methods section.
- Figure 1B. Please list number of mice used in each group in the experiment
- X axis description in Figure 1 C, D are not clear, missing saline group and comparison groups.
- What is the p value for Figure 1D experiment?
- Please list antibodies catalog numbers and concentrations used for both western blot and IHC in the material and methods.
- List correct dose for Figure 3b. Figure legend - PR-619 (15 μM), results text - PR-619 (20 μM)
- Figure 4 is missing a C panel; C panel is listed in figure legend.
- Figure 4 is missing D panel; D panel is listed in results section
- Please provide a better description on how IHC score were calculated; what was the intensity scale used
- For western blots with multiple bands put an arrow pointing to the correct band location (phosphor-JNK, caspase-4)
Reviewer 2 Report
In the manuscript ijms-1413570, Huang et al. investigated the antitumor effect of DUB inhibitor (PR-619) in cisplatin-resistant bladder urothelial carcinoma. The authors showed that PR-619 improved both apoptotic and cytotoxic effects of cisplatin in cisplatin-resistant T24/R cells, which was associated with concurrent suppression of c-Myc expression. In a xenograft nude mouse model, PR-619 significantly improved the antitumor effects of cisplatin. These results highlight the therapeutic target of DUB for efficient treatment of cisplatin-resistant urothelial carcinoma. Overall, this study is well-designed and performed. The methods and results are adequately described. Accordingly, I would recommend the publication of this study after addressing the following minor concerns;
- the animal experiment is not well detailed. What the no of mice? control? grouping?
- Why the authors decided to in vitro test PR-619 at 20uM? please include it in the main MS (better in line 144).
- I suggest that the authors extend their discussion part to cover the most recent studies about Cisplatin and connect these to their findings.
- The authors should discuss, what would be the effect of PR-619 on specific DBU? also what the author think about the effect of specific DBU inhibition on both Cisplatin effect and treatment of bladder cancer?
- Please add a new section for conclusion and outlook. Lines 248-252 are not sufficient.
